# Medication-Overuse Headache: Differences between Daily and Near-Daily Headache Patients

**DOI:** 10.3390/brainsci6030030

**Published:** 2016-08-15

**Authors:** Abouch V. Krymchantowski, Stewart J. Tepper, Carla Jevoux, Marcelo M. Valença

**Affiliations:** 1Headache Center of Rio, Rio de Janeiro 22031-071, Brazil; carlajevoux@uol.com.br; 2Geisel School of Medicine, Dartmouth College, Hanover, NH 03755-1404, USA; sjtepper@gmail.com; 3Federal University of Pernambuco, Recife 52041-040, Brazil; mmvalenca@yahoo.com.br

**Keywords:** medication-overuse headache, daily, near-daily, migraine, chronic migraine, withdrawal

## Abstract

Medication-overuse headache (MOH) is a challenging neurological disease, which brings frustration for sufferers and treating physicians. The patient’s lack of adherence and limited treatment evidence are frequent. The aim of this study was to compare the outcome and treatment strategies between consecutive MOH patients with daily and near-daily headache from a tertiary center. Methods: Every consecutive patient seen between January and December 2014 with the diagnosis of MOH was included. Psychiatric comorbidities, inability to inform baseline headache frequency, current or previous two-month use of preventive medications, and refusal to sign informed consent were exclusion criteria. The patients were evaluated in thorough initial consultations and divided in two groups based on their baseline headache frequency. The diagnosis and treatment strategies were clearly explained. The filling out of a detailed headache diary was requested from all patients. Endpoints compared headache frequency and adherence after two, four, and eight months between the two study groups. Results: One-hundred sixty-eight patients (31 male, 137 female) met the inclusion criteria. Nineteen patients (11.3%) were excluded. All patients had migraine or chronic migraine as primary headaches. Eighty had daily (DH), and 69 near-daily headache (NDH), at baseline consultation. Mean baseline frequency was 24.8 headache days/month (18.9 days/month for the near-daily group), average headache history was 20.6 years and mean time with >15 headache days/month was 4.8 years. Outpatient withdrawal, starting prevention, and enforcing the correct use of rescue therapy was carried out with all patients. After two months, 88% of the DH and 71% of the NDH groups adhered to treatment (*p* = 0.0002). The HF decreased to 12 and 9 headache days/month, respectively in DH and NDH groups (*p* > 0.05, non-significant) (Intention-to-treat (ITT) 14 DH; 12 NDH; *p* > 0.05). After four and eight months, 86.3% and 83.7% of the DH patients, and 59.4% and 55% of the NDH patients were still under treatment (*p* = 0.0003 and *p* = 0.0001). The HF decreased, respectively, to nine and nine headache days/month in the DH patients compared to 6 and 7 headache days/month in the NDH group (*p* > 0.05) (ITT, 12; 12; DH; 10; 11; NDH; *p* > 0.05). Conclusions: Although open studies provide limited conclusions, withdrawing overused medications and starting prevention may have helped the favorable outcomes. However, daily headache patients had a significantly higher adherence and lower relapse rates than near-daily headache patients, despite a considerable reduced headache frequency in both groups. Additionally, real-world patient studies are scarce and the comparison between these two subsets of patients may be useful to guide clinicians in approaching their patients. Controlled studies are necessary to confirm these observations.

## 1. Introduction

Medication overuse headache (MOH) is a disabling disease, affecting nearly 2% of the population [1,2]. It may represent the majority of headache patients seeking care in tertiary centers and its prevalence may be as high as 70% among referred patients. MOH represents a subset of chronic daily headache occurring during or from overuse of one or more classes of headache acute medications [3,4,5]. Although with a high morbidity, scarce evidence is available regarding treatment strategies, which are mostly based on recommendations from expert opinion [6,7]. However, multidisciplinary treatment, absence of psychiatric comorbidities, high motivation, and overuse of drugs other than benzodiazepines, barbiturates, and opioids are favorable outcome factors [7,8].

Real-world patients are consistently excluded from the few available trials on migraine or chronic migraine with medication-overuse headache, and conclusions from these studies may not apply to real-world headache patients seeking care in tertiary centers, especially private centers [9,10]. These patients are often considered difficult or even refractory, but most of the times the lack of a previous comprehensive approach may have been the real reason for past treatment failures [5,7,11].

Current available evidence suggests that outpatient detoxification is enough for a successful withdrawal and for the reduction of headache parameters as well as decreased consumption of overused medications [7,12,13]. However, sustained response based on advice alone is usually difficult [9,12]. In addition, even among patients who succeed in detoxification, nearly one third will relapse within the first year of follow-up [7,8]. It is uncertain whether patients who have a baseline frequency of daily headache adhere better to the treatment strategies and experience a higher reduction of HF in comparison to those who had near-daily headache [8,9,12,13].

The aim of this study was to describe the patient’s characteristics and treatment strategies carried out on consecutive subjects with MOH in a tertiary center. Additionally, it was our objective to discuss these treatment strategies, including preventive medications and outcomes, regarding adherence and headache reduction over the periods of two, four, and eight months for subsets of patients with daily headache and near-daily headache.

## 2. Methods

Consecutive new patients at a private tertiary center in Rio de Janeiro, Brazil, with the diagnosis of Medication Overuse Headache (MOH) according to the International Classification of Headache Disorders (ICHD-3 beta) [14] evaluated between January and December 2014, were included. Inclusion criteria were: (1) 15 or more headache days per month; (2) 18–65 years of age; and (3) primary headache diagnosis of migraine, chronic migraine, or tension-type headache. Exclusion criteria were inability to inform, accurately, baseline headache frequency during the previous six months, presence of major psychiatric comorbidities other than depression and anxiety, use of pharmacological agents for prevention of primary headaches during the previous two months, rejecting participation, and failure to sign the informed consent. Patients who underwent previous attempts of treatment or detoxification were included as well, if inclusion criteria were met.

All patients were evaluated by the same physician (AVK) in initial consultations of no shorter than one hour and before the initial consultation, filled out a detailed questionnaire including a short version of the SCID-II (Structured Clinical Interview for DSM-IV Axis II Disorders), Beck depression inventory, Midas, and questions about the headache characteristics and evolution through the time period. Moreover, during the consultation the patients received a comprehensive approach, explanations regarding the nature of the headache, and details of the treatment. In addition, all received in-depth written material explaining the treatment strategies. A detailed headache calendar to be filled out on a daily basis was presented and requested from all patients. Follow up visits were scheduled for two months, four months, and eight months. The patients were divided in two groups according to the assured baseline headache frequency during, at least, the previous six months. The first group had daily headache and the second group had the presence of headache in four, five, or six days per week, i.e., near-daily headache.

Subjects were patients seeking care for the first time, and after discussion, written informed consent was presented. Its signing was requested for those willing to participate and the study was approved by the ethics committee of the Universidade Federal de Pernambuco (Ethic approval code: CAAE 02090172000-9).

Intention to treat analysis was carried out considering the headache frequency at baseline in the case of the patients who did not return for the two-month visit and the headache frequency at the two-month visit for those not returning at four months, and at the four-month visit for those who did not return at eight months, or the last observation carried out forward approach (LOCF).

### Statistical Analysis

The Kolmogorov-Smirnov test was used to define the distribution type of variables. If numerical variables presented a normal distribution Student *t*-test was used. When variables did not present a normal distribution we used the nonparametric test Mann-Whitney. For categorical variables we applied chi-square test. The data are shown as mean ± standard deviation. A *p*-value < 0.05 was considered significant. The statistical analyses were performed using the Statistical Package Prism (version 5.00 (2007), GraphPad Prism, La Jolla, CA, USA).

## 3. Results

The main characteristics of the study population are in Table 1. One-hundred sixty-eight consecutive patients (31 male, 137 female) met the inclusion criteria between January and December 2014 at the Headache Center of Rio de Janeiro. Eight patients (4.8%) were found to have personality disorders, four (2.4%) patients had taken preventive medications during the previous two months, and seven (4.2%) declined to sign the informed consent. These 19 patients were excluded. One-hundred forty-nine patients (20 male, 129 female), 18–65 years (mean 37.5 ± 11.7) were included in the study. Eighty out of the 149 patients (53.7%) reported previous treatment attempts with other than acute medications. Among those 80, 52 (65%) had daily headache and 28 (35%) had near-daily headache at baseline. In addition, 66 reported no improvement with preventive medications, and 14 reported failing to adhere to medical recommendations or prescriptions.

All patients had migraine (with aura, without aura, or both) or chronic migraine as primary headaches according to the International Classification of Headache Disorders (ICHD-3 beta) (Table 1). The diagnosis of the primary headaches were made based upon the headache history prior to overusing acute medications and not on the features presented during the baseline visit.

The mean baseline frequency was 24.9 ± 5.9 (min 16, max 30) headache days/month. The daily headache group had obviously 30 headache days/month compared to a mean of 18.9 headache days/month in the near-daily headache group (*p* < 0.05). The average headache history was 20.9 ± 11.7 (min 1, max 55) years and time with ≥15 headache days/month was 4.5 ± 5.3 (min 0.5, max 32) years. Both groups had similar and no significant differences regarding headache history. Additionally, all patients were overusing symptomatic medications (SM), and 59/149 (39.6%) were using more than one pharmacological class. No patients were overusing barbiturates or opioids, but 12/149 (8.1%) were using benzodiazepines less than eight days per month, while 3/149 (2.0%) patients were overusing it. The pharmacological classes of medications overused by the patients are presented in Table 2 and there were no significant differences regarding the groups of DH and NDH.

Outpatient withdrawal from overused medications was carried out with all patients, who received different preventive treatment choices starting on the sixth–eighth day and the combination triptans plus a nonsteroidal anti-inflammatory drug (NSAID) for the acute attacks (maximum of 2 days/week) (Table 3). One-hundred one patients out of 149 (67.8%) received prednisone during the first 5–7 days as a bridge to detoxification. Among those who had it, 55/80 (68.8%) of the DH and 46/69 (66.7%) of the NDH patients (*p* > 0.05). The selection of these patients was based on the presence of comorbidities, higher consumption of symptomatic medication, expertise of the treating physician, and degree of tolerance with the headache escalation perceived during the initial consultation and bias of the treating physician. There was no outcome differences between those who received or not received prednisone as well as no differences were encountered between those who used it on five or on seven days.

In addition, there were no significant differences between the groups regarding treatment choices, with the exception of the monotherapeutic use of sodium divalproate, which was used in 11 NDH vs. 1 DH patients (*p* < 0.05) (Table 3). At this time, it was not possible to determine whether those patients having received specific combinations or more preventive medications performed better or presented better outcomes than those who took one or two preventive agents.

After two months, 10/80 (12.5%) of DH and 20/69 (29%) of the NDH patients were lost to follow up. The adherence rate at this time point was 88% vs. 71% (*p* = 0.0002). After four and eight months, respectively, 109 and 105 patients of the whole studied sample were under treatment with a mean headache frequency of 7.6 and 8.3 headache days/month. Sixty eight (85%) of the DH and 41 (59.4%) (*p* = 0.0003) of the NDH patients adhered after four months, whereas 67 (83.8%) of the DH and 38 (55.1%) of the NDH patients were still under treatment after eight months (*p* = 0.0001). The intention to treat (ITT) analysis found that the headache frequency after two months, four months, and eight months was, respectively, 14, 12, and 12 headache days/month for DH patients and 12, 10, and 11 for NDH patients (*p* > 0.05) (Table 4, Figure 1 and Figure 2). After eight months, relapses or the use of symptomatic medications in 10 or higher days per month were observed in 18 patients (26.8%) of the DH and in seven (18.4%) of the NDH groups (*p* < 0.05). However, if we use the ITT analysis, 21 patients (three who were lost to follow-up between month 2 and month 8; 31.3%) in the DH group and 17 (10 who were lost to follow-up between month 2 and month 8; 44.7%; *p* < 0.05) in the NDH were back to the overuse standard.

## 4. Discussion

This was a follow up study describing the particulars of real-world MOH patients attending a tertiary center in a specific geographic area. Main methodological flaws include counting on patient’s recall to assure the baseline headache frequency and the amount of symptomatic medication consumption. The findings may not reflect the patients and approaches carried out in other countries, but do reflect the necessity of a comprehensive approach, with long lasting initial consultations and encouragement of withdrawing from overused medications. Tassorelli et al. published a protocol demonstrating efficacy in treating MOH in different countries, but did not include Brazilian patients [7]. In addition, although there were few differences in conducting the treatment of our sample in relation to the patients of these authors [7], our main line of treatment, for both groups of patients, was: providing information regarding the nature of the pain, withdrawing abruptly, following the patients closely and initiating prevention, and providing for the judicious use of rescue or acute therapy. Our approach is similar to the approach carried out by Munksgaard et al. [8], who studied 86 patients previously unsuccessfully treated by specialists and referred to a specialized tertiary center. The patients were followed during 12 months, and 71 (82.6%) remained out of MOH. These authors reported 42 (48.8%) of their patients achieving ≥50% reduction in headache frequency, which decreased from a mean of 22.3 to respectively 16.7 and 13.7 headache days/month after two and 12 months. We observed a ≥50% headache frequency reduction after two, four, and eight months in most of our patients, even using an intention to treat analysis and having considered, respectively, the 65% of DH and 40.6% of NDH patients with previous treatment failures.

While we were able to detoxify 79.2% of the patients using our treatment strategies (at two months, 88% of DH and 71% of NDH patients, *p* < 0.05), Tassorelli et al. were able to carry out a wean in 85.4% of their 376 enrolled patients [7]. However, we decided to use various preventive treatments in all of the patients, which was not the case in 17.1% of their patients [7].

Although it is not the scope of the present study to discuss why specific preventive medications were chosen for our patients or to claim its effectiveness, since it was chosen based in personal experience and not on available evidence, it is interesting to note that either some of the Tassorelli group’s treating physicians or treated patients decided not to use or take preventive pharmacological agents after the detoxification process, even for daily or near-daily headaches [7]. In addition, the authors did not perceive differences in headache outcome among their patients who received, or not, preventive medications.

Based on our experience and on what is available in the literature for Brazilian patients from tertiary centers, the use of preventive medications in migraineurs with high frequency of headache attacks or in chronic migraineurs is the rule, although evidence regarding combination of pharmacological agents is lacking [15,16,17,18].

Additionally, we were gratified to find a reasonable adherence rate, especially among the daily headache subset of patients (88% after two months and 83.5% after eight months) despite the presence of nearly 54% of our patients having undergone previous treatment failures or simply lack of adherence. We are unable to explain or compare these findings with other groups, because no comparisons were reported for subsets of patients with or without previous unsuccessful treatment attempts in other studies, but Munksgaard et al. were able to reverse medication overuse headache in most of they described as “treatment-resistant patients” [8]. Moreover, failure in previous detoxification procedures or treatment attempts has commonly used an exclusion criterion in some of the published trials, which may make the current series of more practical clinical utility [7].

However, we were able to compare the outcome of the two subsets of patients. While those with daily headache at baseline showed a higher than 80% adherence rate at eight months, the patients with baseline near-daily headache demonstrated a statistically significant lower adherence rate (55%) at this time point (*p* = 0.001).

Our study had the advantage of a patient population, in the two groups, not including opioid or barbiturate overuses, and this often predicts a greater likelihood of success [4,8,19]. Although we have had a few patients overusing benzodiazepines, they were not representative of our sample.

Munksgaard and co-workers [8] demonstrated, with the help of other centers in different countries, that the combination of advice, detoxification, withdrawal, and prevention is a plausible and effective way to improve these patients and to decrease headache frequency. They [8], too, did not have a preponderance of opioid and barbiturate overusers.

Although relapse is common and may be more frequent in those having gone through outpatient versus inpatient detoxification, which was the case with our patients, we did not study outcomes for longer than eight months. Hagen et al. [20] evaluated 61 patients who participated in a randomized open-label prospective study over four years. The authors closely followed 50 of the subjects who reduced the frequency of headache days/month and consumption of symptomatic medications/month and the resultant headache indices/month. Only 32% of their sample was considered as responders, while 34% met criteria for medication overuse headache again after 4 years, that is, showed relapse.

In comparison with our findings, among those 67 DH and 38 NDH patients still under treatment after eight months, the percentage of those relapsing was similar for the DH group (18 patients, 26.8%), but lower for the NDH patients (seven patients, 18.4%), even though we used a stricter standard to consider the overusing of symptomatic medications (10 or more days/month). However, if we use ITT analysis, 21 patients (including three who were lost to follow-up between month 2 and month 8; 31.3%) of the DH subset and 17 (including 10 who were lost to follow-up between month 2 and month 8; 44.7%) in the NDH were back to a medication overuse pattern, that is, relapsed as well.

Interestingly, although we emphasized the education of the patients and dealt exclusively with patients from a private tertiary center, supposedly possessing higher educational standards and, therefore, motivation, nearly 30% of the patients failed to return for the follow-up visit at the eighth month (third follow-up visit) [15]. These numbers are quite similar to those 31.7% of the population of patients studied by Tassorelli and colleagues [7] from various centers in Europe and South America, but not Brazil. However, a comparison between the subsets of DH patients and NDH patients showed considerable differences. While most of the daily headache patients (67; 83.7%) maintained and followed the treatment orientations after eight months, only half of the near-daily patients were still adhering after this timeframe (38; 55%; *p* = 0.0001). Thus, 16.3% of DH and 45% of NDH patients were lost. One might consider not only economic issues for losing patients to follow up, since payments for each visit are necessary, but, in addition, baseline degree of burden and incapacitation suggesting that patients formerly experiencing more headache days adhered better than those having head pain fewer days a week [15,21,22].

In the intention to treat analysis (ITT), headache frequency was reduced from 30 headache days per month to 12, 9, and 9 headache days per month, respectively after 2, 4, and 8 months of treatment in the DH group. Regarding NDH patients, headache days per month were reduced from an average of 19 headache days/month to, respectively, 9, 6, and 7 headache days/month. Although the difference in headache frequency was significant during baseline and after the three endpoints (*p* < 0.001) compared to baseline in both subsets of patients, the comparisons between headache frequencies after 2, 4, and 8 months were not different regarding these two groups (*p* > 0.05).

One possible explanation for the favorable outcome and adherence is the fact that the studied subjects were private patients from a tertiary center, usually implying, at least in Brazil, better socioeconomic status and, perhaps, a higher motivation [15]. A potential selection bias could be due to the higher socioeconomic status and more motivated patient sample. This hypothesis was also suggested by Tassorelli et al. [7]. In fact, as in their study, out patients completed detoxification at home and were likely to comply with treatment strategies. Additionally, as demonstrated by our patients, previous failures of detoxification cannot rule out the prospect of success in a further wean from a different center with different protocols and approaches [7].

The approach for our patients differed from many of the recent MOH studies. Our bridge therapy for detoxification was oral prednisone instead of anti-dopaminergic agents, including metoclopramide, which we avoided due to the potential for drowsiness [4]. Prednisone was chosen based on the expertise of one of the authors (AVK) and mostly for those patients with a higher consumption of symptomatic medication. However, no significant differences were encountered regarding adherence and amelioration between those having used prednisone and those not having done that.

A drawback to this study was the absence of a control group or alternate treatment pathways, as well as comparison between treatment choices among the two subsets of patients. One of the authors (AVK) previously published a randomized, controlled comparative effectiveness study on three treatment paradigms for MOH [18]. This case series was meant to evaluate patients who had, for the most part, tried treatments without success to see if an additional attempt might yield better outcomes. However, we still do not know whether the simple detoxification without prophylaxis would have provided improvement as well, in addition to the fact that the one of the preventive treatment choices cannot be assured as more efficient or better than another.

## 5. Conclusions

We describe a prospective case series of two groups of patients with medication overuse headache in a tertiary Brazilian headache center. One-hundred forty-nine patients were enrolled in an individualized treatment program which included wean of overused medication, bridging medications to help with withdrawal, and intense counseling to improve adherence. This sample was further divided into a group with daily headache and another one with near-daily headache at baseline.

Baseline headache frequency was 30 or 18.9 headache days per month. Eighty patients (53.7%), being 52 (65%) in the DH group and 28 (35%) in the near-daily group reported previous treatment attempts with other than acute medications. Among those, 66 reported no improvement with preventive medications, and 14 reported failing to adhere to medical recommendations or prescriptions. All patients had migraine (with aura, without aura, or both) or chronic migraine as primary headaches according to the International Classification of Headache Disorders (ICHD-3 beta).

Bridging medication, prednisone, was used in 64.8% of patients (68.7% for DH and 66.7% for NDH patients). All patients received individualized preventive treatment and triptan plus NSAID for acute attacks (maximum of 2 days/week).

After 4 months, 68 daily headache and 41 near-daily headache were under treatment and after eight months, 67 DH and 38 NDH patients were under treatment, with a mean headache frequency, respectively, of 9 and 6, and 9 and 7 headache days/month.

Thus, a majority of patients showed dramatic reduction in headache days, restoration of episodic headache pattern, and good adherence with conventional, personalized, specialized headache care, despite previous therapeutic failures. Careful evaluation, education, and follow-up with patients previously felt to be refractory, can result in optimal clinical outcomes. Future controlled studies comparing the effect of the different prophylactic regimens used here, as well as comparison against detoxification without prophylactics, are needed to confirm these observations.

## Figures and Tables

**Figure 1 brainsci-06-00030-f001:**
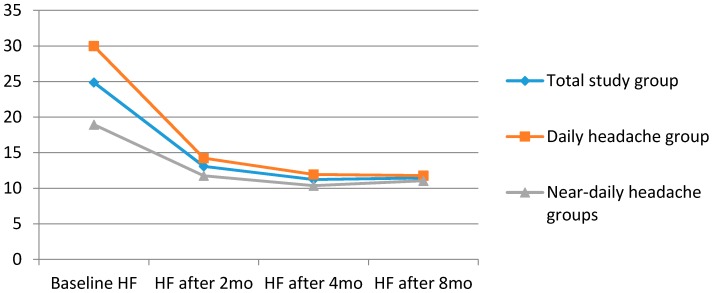
Headache frequency (HF, number of days with headache/30 days) across time periods evaluated (Intention to treat). Groups of patients were divided by headache frequency at baseline. *n* = 149 (total study group). *n* = 80 (daily headache group) *n* = 69 (near-daily headache group).

**Figure 2 brainsci-06-00030-f002:**
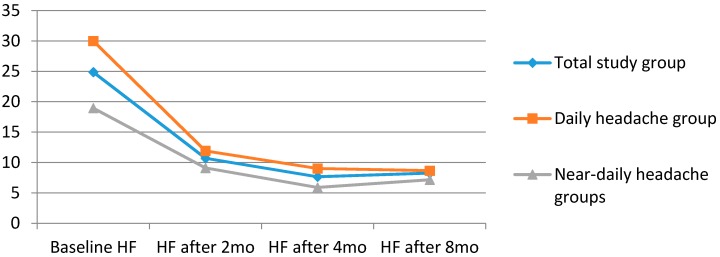
Headache frequency (HF, number of days with headache/30 days) across time periods evaluated considering patients who adhered to the treatment. Groups of patients were divided by headache frequency at baseline. *n* = 149 (total study group); *n* = 80 (daily headache group) and *n* = 69 (near-daily headache group).

**Table 1 brainsci-06-00030-t001:** Baseline characteristics of the 149 patients who entered the study.

*n*	168	
Patients included	149	
Age in years (mean ± SD)	37.5 ± 9	
Gender	Male	Female
	20	29
Daily Headache Patients	80 (53.7%)	
Near-Daily Headache Patients	69 (46.3%)	
Duration in years of headache (mean ± SD)	20.9 ± 11.7	
Duration in years of headache > 15 days/month (mean)	0.5–32 (4.5)	
Mean days of headache/month	24.9	
Mean days of headache/month (NDH patients)	18.9	
Days of symptomatic medication overuse/month	22.6	
Patients with previous treatment attempts	80	
Daily headache patients	52 (65%)	
Near-daily headache patients	28 (35%)	
Primary headache diagnosis		
Migraine Without aura	131 (72 DH; 59 NDH)	
Migraine with and without aura	15 (8 DH; 7 NDH)	
Chronic Migraine	3 (3 NDH)	
Tension-Type Headache	0	

**Table 2 brainsci-06-00030-t002:** Number of patients overusing symptomatic or acute medications by pharmacological class.

Simple analgesics	20 (7 DH; 13 NDH)
Combination analgesics with caffeine	44 (21 DH; 23 NDH)
Triptans	65 (33 DH; 32 NDH)
Combination analgesics with ergots	17 (11 DH; 6 NDH)
Benzodiazepines	3 (1 DH; 2 NDH)
Opioids	0
Barbiturates	0
More than one class	59 (31 DH; 28 NDH)

**Table 3 brainsci-06-00030-t003:** Preventive medications prescribed to the included patients in both groups “daily headaches” and “near-daily headaches”.

Preventive Treatment Choices	*n* = 149	Daily Headache	Near-Daily Headache	*p* *	Dose Range
		*n* = 80	*n* = 69		
Monotherapy	12 (8%)	1 (1.3%)	11 (16%)		
Two preventive medications	41 (27.5%)	23 (28.7%)	18 (26%)		
Three preventive medications	64 (43%)	34 (42.5%)	30 (43.5%)		
Four preventive medications	32 (21.5%)	22 (27.5%)	10 (14.5%)		
Sodium Divalproate	12	1	11	0.0010	500–750 mg/day
Sodium Divalproate + Topiramate	8	2	6	0.0943	500 mg + 100–150 mg/day
Topiramate + Nortriptyline	14	6	8	0.3931	100–150 mg/day + 20–30 mg/day
(Nortriptyline + Tizanidine + Flunarizine) †	46	24	22	0.8040	20 mg + 8–12 mg + 2–3 mg/day
(Nortriptyline + Tizanidine) †	10	8	2	0.0841	20 mg + 8–12 mg/day
Sodium Divalproate + (Nortriptyline + Tizanidine) †	16	8	8	0.7540	500 mg + 20 mg + 8–12 mg/day
Sodium Divalproate + Nortriptyline	4	4	0	0.0598	500 mg + 20 mg/day
Sodium Divalproate + Candesartan	4	3	1	0.3863	500 mg + 8–16 mg/day
(Nortriptyline + Tizanidine + Flunarizine + Pizotifen) †	14	10	4	0.1620	20 mg + 8–12 mg + 2–3 mg + 0.8–1.2 mg/day
Topiramate + (Nortriptyline + Tizanidine + Flunarizine) †	18	12	6	0.2390	100–150 mg + 20 mg + 8–12 mg + 2–3 mg/day
Topiramate + (Nortriptyline + Tizanidine) †	2	2	0	0.1861	100–150 mg + 20 mg + 8–12 mg/day
Topiramate + Candesartan	1	0	1	0.2800	100–150 mg + 8–16 mg/day

* Chi-square test; † Compounded in the same capsule (posology = once a day).

**Table 4 brainsci-06-00030-t004:** Adherence and outcome of the studied patients divided by headache frequency at baseline in both groups “daily headache” (*n* = 80) and “near-daily headache” (*n* = 69).

	Daily Headache	Near-Daily Headache	*p*	OR (95% CI)
**Previous treatment Failures**				
	52/80 (65%)	28/69 (41%)	0.0052 *	2.6 (1.3–5.1)
**Number of days with headache/30 days**				
ITT	30	19 ± 3	<0.001	
After 2 months	14 ± 10	12 ± 8	>0.05	
After 4 months	12 ± 9	10 ± 8	>0.05	
After 8 months	12 ± 9	11 ± 13	>0.05	
**Baseline**				
After 2 months	12 ± 9	9 ± 8	>0.05	
After 4 months	9 ± 6	6 ± 5	>0.05	
After 8 months	9 ± 5	7 ± 6	>0.05	
**Adherence to treatment**				
After 2 months	70/80 (88%)	71% (49)	0.0002 *	4.3 (1.9–9.6)
After 4 months	69/80 (86%)	59.4% (41)	0.0003 *	4.3 (1.9–9.5)
After 8 months	68/80 (85%)	56.5% (39)	0.0001 *	5.8 (2.7–12.4)

* Chi square; ITT, Intention-To-Treat.

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
