# Peer review of "Medication-Overuse Headache: Differences between Daily and Near-Daily Headache Patients"

_brainsci, 2016, doi:10.3390/brainsci6030030_

Round 1

Reviewer 1 Report

The authors report an open real world follow-up study of patients with migraine and medication overuse. Patients were divided in two groups: daily headache and nearly daily headache. Both groups are comparable regarding demographic baseline characteristics. Both groups showed benefit from the treatment strategies of the first author showing more success in patients suffering from daily headache compared to those with nearly daily headache.

However, some points could be discussed with the manuscript:

1.     Do the authors have a hypothesis what might be the difference between patients suffering from daily compared to nearly daily headache? (please present the rationale for the study in more details)

2.     Daily headache means the patient report no headache free day within (minimum) the last three months? Nearly daily headache reaches from a minimum of 15 headache days per month to at least one headache free day per month? (now in the manuscript: “…the presence of headache in four, five or six days per week” – usually headache frequency is reported per month). Please clarify.

3.     Table 1 and 2 could be organized in a different manner and (it’s the authors decision) integrated into one single table. Especially table 1 is a little bit confusing and might be much easier to understand if the authors present in a different way: Item   - All patients – Daily headache – Nearly daily headache.

4.     Which dose of prednisone was used for treatment of MOH during detoxification?

5.     Minor:

Spelling mistakes: table 1: Migraine Without aura (W)

Reference 10 and 19 are the same.  

Author Response

Reviewer 1

1.     Do the authors have a hypothesis what might be the difference between patients suffering from daily compared to nearly daily headache? (please present the rationale for the study in more details)

We have added to the rationale to the introduction, page 3.

2.     Daily headache means the patient report no headache free day within (minimum) the last three months? Nearly daily headache reaches from a minimum of 15 headache days per month to at least one headache free day per month? (now in the manuscript: “…the presence of headache in four, five or six days per week” – usually headache frequency is reported per month). Please clarify.

We have rewritten the relevant sentences to clarify this matter, in the methods section, page 4:

3.     Table 1 and 2 could be organized in a different manner and (it’s the authors decision) integrated into one single table. Especially table 1 is a little bit confusing and might be much easier to understand if the authors present in a different way: Item   - All patients – Daily headache – Nearly daily headache.

Changes were made as suggested, page 7.

4.     Which dose of prednisone was used for treatment of MOH during detoxification?

Added in Methods, page 4. Just to emphasize: For those having received it during 7 days, it was 60mg/day for 3 days; 40mg/day for 3 days and 20mg/day for 1 day. For those having used during 5 days, it was 60mg/day for 3 days and 40mg/day for 2 days.

5.     Spelling mistakes: table 1: Migraine Without aura (W)

Corrected

6.     Reference 10 and 19 are the same.

Reference 19 was deleted.

Reviewer 2 Report

Review:  MOH:  Differences between DH and NDH Patients

This is an interesting study which deals with MOH induced by analgesics, triptans, ergots and benzodiazepines, or combinations of the above.  Of note,  no subjects were entered for which barbiturates or opioids were entered.  These are major medication overuse headache inducing agents (MOHA) in other parts of the world.  The treatment paradigm consisted of sudden withdrawal of the MOHA and then preventative antimigraine medications (PAM) including tricyclics, antiepileptics and flunarizine with occasional use of tizanidine and candesartan.

It is not clear if the patients were given a symptomatic medication to treat severe headaches after the treatment protocol was begun.

Questions which arise are as follows:

The treatment protocol is presented only scantily.  More information is needed as to doses of PAM and the amount of symptomatic medication needed or allowed.

Bridging with steroids was carried out.  Were there complications from this , and did it represent an improvement over the anti-dopamine agents.

How much of the MOHA was being used before withdrawal took place?

DId all or only some patients have very severe headaches on withdrawal?  Did this depend on the extent to which the MOHA was being used?

Were the patients allowed to use the MOHA  again after its withdrawal?  If so, how often and how much?

Was re-use of the MOHA instrumental in relapse or failure to follow-up in the study.

Patients with chronic Migraine typically have days with “big” headaches and days with “little” headaches.  The days with “big”  headaches are the ones that are most disabling and the number of “big” headaches /month is usually an indicator of the degree of success of the program and whether the patient can hold a job or function well in domestic responsibilities.  Is there any information about days with severe headaches and days with mild headaches in the total headache days reported.

How many of the subjects were able to either continue in their job or maintain employment after 2, 4, 6, or 8 months in the program?

The finding of better outcome for DH as opposed to NDH subjects is interesting.  Are there any clues to help determine why this occurred?

As with other “real life” reports there is a great disparity of the data available .  the questions above are posed to help clarify some of the disparity.

Author Response

Reviewer 3

1.     The treatment protocol is presented only scantily.  More information is needed as to doses of PAM and the amount of symptomatic medication needed or allowed.

This was included in the methods section.

2.     Bridging with steroids was carried out.  Were there complications from this, and did it represent an improvement over the anti-dopamine agents.

We did not use anti-dopaminergic agents. We chose steroids since it is our standard approach for the last 25 years. We believe anti-dopaminergic agents promote incapacitating drowsiness and, therefore, don’t use that. Nothing related to side effects of this short cicle of steroids was noted

3.     How much of the MOHA was being used before withdrawal took place?

The data in question were included in the results section.

4.     Did all or only some patients have very severe headaches on withdrawal?  Did this depend on the extent to which the MOHA was being used?

This was not the objective of the study, but it may be observed in future studies. With the data gathered for the study, it was not possible to determine that

5.     Were the patients allowed to use the MOHA again after its withdrawal?  If so, how often and how much?

This was in the methods section.

6.     Was re-use of the MOHA instrumental in relapse or failure to follow-up in the study.

We cannot answer this with our data.

7.     Patients with chronic Migraine typically have days with “big” headaches and days with “little” headaches.  The days with “big”  headaches are the ones that are most disabling and the number of “big” headaches /month is usually an indicator of the degree of success of the program and whether the patient can hold a job or function well in domestic responsibilities.  Is there any information about days with severe headaches and days with mild headaches in the total headache days reported.

This was not addressed in the present study.

8.     How many of the subjects were able to either continue in their job or maintain employment after 2, 4, 6, or 8 months in the program?

This was not addressed in the present study.

9.     The finding of better outcome for DH as opposed to NDH subjects is interesting.  Are there any clues to help determine why this occurred?

The outcome was similar in both groups of patients regarding a decreased number of days with headache (60%, 70% and 70% in DH vs. 53%, 68% and 63% in NDH, respectively, at 4, 6 and 9 month of treatment), but the adherence rate was higher in DH group. After eight months, relapses or the use of symptomatic medications in 10 or higher days per month were observed in 18 (26.8%) patients of the DH and in 7 (18.4%) of the NDH groups (p<0.05). However, if we use the ITT analysis, 21 patients (31.3%) in the DH group and 17 (44.7%; p<0.05) in the NDH were back to the overuse standard. Thus, on the contrary, in respect to this parameter (i.e. relapses) there was a better outcome for NDH as opposed to DH subjects. (This paragraph was added to the conclusion.)

We thank you all for your kind attention and look forward to hearing from you in due cause.

Reviewer 3 Report

This article focuses on out-patient treatment of 149 Medication-overuse headache patients recruited consecutively at a private tertiary headache clinic. They all had migrainous headache (even though tension-type headache could, according to the protocol, be included as well). Main overused medication groups were triptans, more than one drug class, combinations with caffeine, simple analgesics and ergot combinations. All were treated with withdrawal using prednisone bridging therapy for 5-7 days for the majority (68%). All were also give preventive medication using 12 different treatment regimens including the following medications: divalproate, topiramate, nortriptyline, tizanidine, flunarizine, candesartan and pizotifen. Follow-up time was to 8 months. Though main outcome variable was not specified in the protocol it seems that this was headache frequency (days per month). On-going medication overuse/relapse was assessed as proportion with symptomatic medication >10 days per month. Patients were divided into two groups: those with daily headache (80 patients) and those with near-daily headache (69 patients with headache 4-6 days per week).

It seems the study is not reported in any clinical trials registries (at least not in Clinical trials or EUDRA-CT).   

The theme and pragmatic focus are commendable. Treatment practices may, as the authors also suggest, represent practices specific to Brazil and to the patient selection which follows from a self-paying system.

However, I have a number of serious concerns which, to my mind preclude the publication of the article in it's present form.

Major concerns:

1. Treatment study including several un-documented preventive medication combinations with out the study being registered in clinical trial registries.  

2. Indication of severe patient selection bias for inclusion of only migrainous patients without this being properly indicated. The title should perhaps better state "Migraine with medication-overuse" or "Medication-overuse with migraine/migrainous features" as no patients other than those with co-diagnosis of migraine are included.

3. Treatment regimes and procedures are insufficiently described in Methods. (For example, the preventive medications used are presented only as results - Table 3). Was "Cold turkey" detox (i.e. sudden termination) used for all medication groups or a slow tapering for some? 

4. Outcomes/main outcomes should be presented under Methods/Statistical analysis.

5. Statistical analyses are insufficient (e.g. no mentioning of corrections of p-limits based on multiple testing, no use of adjusted statistics - at least adjusting for age, gender and main overused medication group should be done. There is also no mentioning of taking time- series associations into account in statistical calculations of follow-up data.

6. There are several bias problems which are not sufficiently taken into account - the first has already been mentioned - bias towards migraine patients even though the title seems to state otherwise. Another, perhaps more serious bias is that of different use of preventive regimes in the two groups. For example divalproate monotherapy is almost exclusively used in the near-daily group (11 vs 1 patient), nortriptyline is used for 74 patients in the daily group but only 50 in the near-daily group. This, as well as the high lost-to follow up rate which also seemed to differ between the two groups which are then subsequently compared, may suggest a biased lost to follow-up dependant perhaps on medication side effects as well as varying efficacies of the different regimes. Were there different dropout rates between the different preventive regimes?

7. Why was not a straight forward detox group used as one of the treatment regimes? 

8. Even though this paper is presented as a pragmatic study illustrating praxis at the clinic in question, it seems as though the various medication combinations are chosen almost randomly and it is not stated based on what these treatments were chosen. It is, for example, not clear whether the "different preventive treatment choices" were choices for the treating physician or whether the patient also had a say. 

Minor:

1. Redundancy of figures/tables - figs 1-2 present the same data as in table 4.  

2. Table 4 is difficult to read: presumably the follow up data in the middle should be for ITT and per protocol and there is missing info regarding baseline data for the second. The lower part of the same table should have the same variable results format in both groups - daily headaches is now presented as frequency n/N (%) while near-daily is presented as % (n).

3. It is not clear how many patients achieved primary detoxification from overuse and therefore how many relapsed from this versus how many were never detoxified (bottom page 5). 

4. The Conclusion is not a conclusion but rather formulated as an extra abstract.

5. Regarding referencing, several recent detox studies and reviews have not been included in the reference list. 

Author Response

Reviewer 2

1.     It seems the study is not reported in any clinical trials registries (at least not in Clinical trials or EUDRA-CT).   

Indeed it was not registered, as explained above, due to the different choice of drug(s) considered for use in each case. This is a study showing the real-world scenario of a Brazilian physician, using well-established and traditional prophylactic drugs used by migraineurs. In Brazil, there is no need to do so and only studies with drugs not yet approved and available by our Federal agency (ANVISA), require registration of trials involving its use.

2.     Treatment study including several un-documented preventive medication combinations without the study being registered in clinical trial registries.

The reviewer is correct regarding the lack of studies employing combinations of migraine preventive drugs. The authors believe, at least in part, that this occurs due to a conflict of interest between rival pharmaceutical companies, which have no interest in investing in such studies.   We do believe that our study is an important one with a view to motivating further studies with different combinations of drugs. (This comment was added in the discussion.). In addition, we believe that controlled studies using drug combinations should be performed in traditional research centers and not with paying patients from a private tertiary center. Therefore, we only described what was done without suggesting that some approaches were better than others 

3.     Indication of severe patient selection bias for inclusion of only migrainous patients without this being properly indicated. The title should perhaps better state "Migraine with medication-overuse" or "Medication-overuse with migraine/migrainous features" as no patients other than those with co-diagnosis of migraine are included.

The title was modified as suggested to: “Migraine with medication-overuse: differences between daily and near-daily headache patients”. Just a clarification: At least in Brazil, tertiary centers rarely receive tension-type headache sufferers or pure chronic migraine without medication overuse headache. It may be more common to find so in public centers (please see reference 15) where, sometimes, patients seeking for official work dispensation request medical consultations even with mild headaches

4.     Treatment regimens and procedures are insufficiently described in Methods. (For example, the preventive medications used are presented only as results - Table 3). Was "Cold turkey" detox (i.e. sudden termination) used for all medication groups or a slow tapering for some?

The authors have added better descriptions of the way drugs were used in the methods section and that the symptomatic medications were abruptly suspended. Additionally, when prevention was started and the bridging with prednisone  

5.     Outcomes/main outcomes should be presented under Methods/Statistical analysis.

Accepted. The sentence “Adherence to the treatment and headache reduction over the periods of two, four, and eight months for subsets of patients with daily headache and near-daily headache were the main outcomes.” was added in the Methods Section.

6.     Statistical analyses are insufficient (e.g. no mentioning of corrections of p-limits based on multiple testing, no use of adjusted statistics - at least adjusting for age, gender and main overused medication group should be done. There is also no mentioning of taking time- series associations into account in statistical calculations of follow-up data.

No statistical differences were found in age and sex between the two groups. It was added in the results section and tables

7.     There are several bias problems which are not sufficiently taken into account - the first has already been mentioned - bias towards migraine patients even though the title seems to state otherwise. Another, perhaps more serious bias is that of different use of preventive regimes in the two groups. For example divalproate monotherapy is almost exclusively used in the near-daily group (11 vs 1 patient), nortriptyline is used for 74 patients in the daily group but only 50 in the near-daily group. This, as well as the high lost-to follow up rate which also seemed to differ between the two groups which are then subsequently compared, may suggest a biased lost to follow-up dependant perhaps on medication side effects as well as varying efficacies of the different regimes. Were there different dropout rates between the different preventive regimes?

Tertiary private centers rarely receive Tension-type headache patients or pure chronic migraineurs (without medication overuse headache) at least in Brazil. That’s way most of the patients had migraine before starting medication overuse headache. The statistical analysis was not able to detect any differences between the different regimens of treatment, which may be due to the small number of cases in each type of regimen. In addition, because this study was purely a descriptive particular way of treating patients by an experienced specialist, different approaches like the use of monotherapy with divalproate were probably because the group who received that specific treatment had either less impact of the headache and/or lack of previous treatments and/or less intensity or duration of acute headache medications overuse. On the other hand, those who received a prescription of four different substances compounded in one capsule probably reported, during the first consultation, previous issues with adherence when were asked to take multiple doses or regimens during previous treatments. Because we didn’t aim at discussing efficacy of one agent compared to another in an uncontrolled design like we performed, we just described without concluding the possible effectiveness of the preventive treatments we have used.

8.     Indeed, what we believe as more relevant was the high lost-to follow up rate which also seemed to differ between the two groups which are then subsequently compared. Although it may suggest a biased lost to follow-up dependant perhaps on medication side effects as well as varying efficacies of the different regimes. Were there different dropout rates between the different preventive regimes?

We were unable to determine that with data acquired. We are sorry for that incompetence! We added some details regarding to the choice of different treatment options to the discussion section

9.     Why was not a straight forward detox group used as one of the treatment regimes? 

This was not the objective of this study, but it is a good suggestion for future studies. Anyway, the authors believe that instituting a regimen with preventive migraine drugs after withdrawal of overused medications, is better than merely suspending the symptomatic medication. Moreover, we don’t think it is possible to do that with regular paying patients, since we particularly don’t believe it is very useful

10.  Even though this paper is presented as a pragmatic study illustrating praxis at the clinic in question, it seems as though the various medication combinations are chosen almost randomly and it is not stated based on what these treatments were chosen. It is, for example, not clear whether the "different preventive treatment choices" were choices for the treating physician or whether the patient also had a say. 

This was explained above. It was not a random choice. Instead, we performed what we believed was better for each different patient, its past and/or previous treatment attempts or failures and their burden with the headache. Whenever possible, comorbid conditions were identified and combinations were chosen.

11.  Redundancy of figures/tables - figs 1-2 present the same data as in table 4.  

The redundant figures were deleted.

12.  Table 4 is difficult to read: presumably the follow up data in the middle should be for ITT and per protocol and there is missing info regarding baseline data for the second. The lower part of the same table should have the same variable results format in both groups - daily headaches is now presented as frequency n/N (%) while near-daily is presented as % (n).

Table 4 was improved.

13.  It is not clear how many patients achieved primary detoxification from overuse and therefore how many relapsed from this versus how many were never detoxified (bottom page 5).

This was not possible to accurately determine since a number of patients did not return for follow-up. There was no outcome differences between those who received or not received prednisone as well as no differences were encountered between those who used it on 5 or on 7 days. In addition, we calculated the HF decrease using the ITT in addition to per-protocol and supposed, for this calculation, that these patients were never able to detoxify and reduce the HF

14.  The Conclusion is not a conclusion but rather formulated as an extra abstract.

Several redundant sentences were deleted.

15.  Regarding referencing, several recent detox studies and reviews have not been included in the reference list. 

We added a reference.

Round 2

Reviewer 3 Report

The authors have responded adequately to most of my and the other referee's comments. They have been open about their rationale and methodology and clarified the general objective and limitations with the study.  I do think that the study contains important information and results which ought to be published. I have only two comments which I think the authors need to consider and add.

 I previously asked for a more expanded methodology section considering the treatment regimens used - some additional information has been added such as the  clarification of the abrupt detox and the steroid bridging therapy. However,  the authors still need to add the definite prophylactic treatment regimens used - medications, preparations, dosages etc for each of the treatment regimens used. This is necessary in order to give the reader a possibility to evaluate and possibly to reproduce the results presented. At the very least this could be added as an appendix table. 

This is a pragmatic study which illustrates the practices of the authors and the authors are now more open in describing that they believe direct prophylactic treatment to be better. In addition, they now in their responses state that detox without direct prophylactics is something which should be further studied. They also state that they think that one reason why such studies are important and not done in a more large scale  and RCT like manner is that pharmaceutical  companies have no interest in such studies. Exactly for  this reason it is also important to state clearly that the effect of straight forward detoxification per se, without additional prophylactics should  be the subject of further studies. A sentence under conclusions stating this clearly (e.g. something like "future controlled studies comparing the effect of the different prophylactic regimens used here as well as comparison against detoxification without prophylactics are needed"), should be added.

Author Response

Thank you for the comments. We really agree that explaining the doses and how the treatment was performed sheds light to the review. A new table 3 was added and shows the doses of the medications. The inclusion of the sentence related to the need for further studies proving that the detoxification alone is not as good as the addition of the prophylaxis was added at the end of the discussion and the conclusions.